# Observational retrospective study calculating health service costs of patients receiving surgery for chronic rhinosinusitis in England, using linked patient-level primary and secondary care electronic data

Caroline S Clarke ,[1] Elizabeth Williamson,[2,3] Spiros Denaxas,[3,4] James R Carpenter,[2,5] Mike Thomas,[6] Helen Blackshaw,[7] Anne G M Schilder,[8,9] Carl M Philpott ,[10,11] Claire Hopkins ,[12] Stephen Morris ,[13] On behalf of the MACRO programme team

**Correspondence to**
Dr Caroline S Clarke;
caroline.clarke@ucl.ac.uk

## ABSTRACT

**Objectives** Chronic rhinosinusitis (CRS) symptoms are experienced by an estimated 11% of UK adults, and symptoms have major impacts on quality of life. Data from UK and elsewhere suggest high economic burden of CRS, but detailed cost information and economic analyses regarding surgical pathway are lacking. This paper estimates healthcare costs for patients receiving surgery for CRS in England.

**Design** Observational retrospective study examining cost of healthcare of patients receiving CRS surgery.

**Setting** Linked electronic health records from the Clinical Practice Research Datalink, Hospital Episode Statistics and Office for National Statistics databases in England.

**Participants** A phenotyping algorithm using medical ontology terms identified 'definite' CRS cases who received CRS surgery. Patients were registered with a general practice in England. Data covered the period 1997–2016. A cohort of 13 462 patients had received surgery for CRS, with 9056 (67%) having confirmed nasal polyps.

**Outcome measures** Information was extracted on numbers and types of primary care prescriptions and consultations, and inpatient and outpatient hospital investigations and procedures. Resource use was costed using published sources.

**Results** Total National Health Service costs in CRS surgery patients were £2173 over 1 year including surgery. Total costs per person-quarter were £1983 in the quarter containing surgery, mostly comprising surgical inpatient care costs (£1902), and around £60 per person-quarter in the 2 years before and after surgery, of which half were outpatient costs. Outpatient and primary care costs were low compared with the peak in inpatient costs at surgery. The highest outpatient expenditure was on CT scans, peaking in the quarter preceding surgery.

**Conclusions** We present the first study of costs to the English healthcare system for patients receiving surgery for CRS. The total aggregate costs provide a further

## Strengths and limitations of this study

► Using linked patient-level primary and secondary healthcare records covering 8% of the England population, we provide a comprehensive picture of the costs to the national healthcare system for chronic rhinosinusitis (CRS) surgical patients undergoing surgery for their CRS.

► Our work addresses a paucity of evidence regarding the direct costs of the surgical treatment pathway for CRS in England, and provides a valuable resource to aid commissioning decisions and future research involving surgical treatments for CRS in the UK.

► Coding limitations common in observational data mean that the 'unknown-polyps' subgroup cannot definitively be stated to contain only patients with CRS without nasal polyps as some patients with polyps might also be present if their polyps were not recorded in a standard way.

impetus for trials to evaluate the relative benefit of surgical intervention.

## INTRODUCTION

Chronic rhinosinusitis (CRS) represents a common source of ill health, affecting 5%–12% of the general population.[1] In the UK, 11% of adults reported having CRS symptoms.[2] Symptoms, often poorly controlled,[3] and including nasal obstruction, nasal discharge, facial pain, anosmia and sleep disturbance, have major impacts on quality of life (QOL), possibly greater than the QOL impacts of chronic respiratory disease or angina.[4] In addition, expenditure on rhinosinusitis treatments has been estimated in the

USA as higher than for diseases such as ulcer disease, acute asthma and hay fever.[5] The socioeconomic cost of CRS is significant with 57% of patients reporting absenteeism in Sweden in 2008–2009,[6] 28% experiencing associated anxiety and depression (UK, data collected 2007–2013)[7] and an estimated 19 missed workdays per patient with CRS per year (England, recruitment 2013–2015).[8] In 2011, CRS cost the US healthcare system $8.6 billion with significant direct and indirect costs.[9 10] Our recent systematic review of literature regarding the cost-effectiveness of surgical intervention confirms the lack of UK perspective economic evaluations, particularly relating to the UK healthcare system.[11]

This study forms part of the MACRO programme, 'Defining best Management for Adults with Chronic RhinOsinusitis', and information from this cost analysis will supplement the analysis of the MACRO randomised controlled trial (RCT), which began recruitment in 2018.[12 13] The overarching aims of MACRO are to address major deficiencies in the evidence base for CRS management, establish best practice for management of adults with CRS and design the ideal patient pathway across primary and secondary care. This observational cohort analysis of CRS surgery patients established the costs to the National Health Service (NHS) of treatments received by these patients from general practices/general practitioners (GPs) and in NHS hospitals in England as inpatients (including day cases) and outpatients (OP), and estimated how much they cost, by polyp-defined subgroup as described below, using linked patient-level primary and secondary care electronic health record (EHR) data and mortality data from the Office for National Statistics (ONS). The total aggregated costs to the NHS provide a further impetus for trials to evaluate the benefit of surgical intervention.

## METHODS

### Study design and population

Linked EHRs from the Clinical Practice Research Datalink (CPRD, primary care, covers ~8% of England population),[14] Hospital Episode Statistics (HES, covering inpatient and OP care provided in NHS hospitals in England) and ONS (mortality data) databases were used. Data and phenotyping algorithms were accessed as part of the CALIBER resource.[15 16]

The population used was a subset of the cohort used in previous work by this group that considered the risk of mortality and cardiovascular events following macrolide prescription in patients with CRS.[17] An EHR phenotyping algorithm, comprising primary care and secondary care diagnoses and secondary care procedures deemed to indicate a 'definite' diagnosis of CRS, was developed in collaboration with clinicians (see online supplemental material, section A) using a similar approach to that published by Rudmik *et al*, Lui and Rudmik, and Macdonald *et al*.[18–20] Patients with one or more of these diagnoses or procedures recorded were classified as 'definite' CRS cases,

with the date of diagnosis taken as the date of the first such specified diagnosis or procedure. A further list of 'definite' and 'very likely' surgery OPCS Classification of Interventions and Procedures version 4 (OPCS-4) codes was similarly developed, and the surgical cohort used in this cost analysis was the group of patients with 'definite' CRS who had had surgery defined as either 'definitely' or 'very likely' to have been for CRS (see online supplemental material, section A).

Eligible patients entered the analysis cohort on the latest of current general practice registration date of the patient, date on which research quality data were first provided by the general practice (based on internal CPRD algorithm[14]), their 16th birthday or study start date (1 April 1997). Cases were required to have a minimum of 1-year research quality information prior to their CRS diagnosis, and a minimum of 1-day research quality data at an individual level following diagnosis. Patients left the cohort on the earliest of transfer-out date from the general practice, last data collection from general practice, 80th birthday, death (recorded in either CPRD or ONS) or study end date (29 February 2016). OP data were available from 1 April 2003.

A patient's follow-up period began on their CRS diagnosis date and ended when they left the cohort. The index date around which patients' treatment information was centred was the date on which the first CRS-specific surgery took place during the analysis period, meaning that day 0 could correspond to any calendar date between 1 April 1997 and 29 February 2016 for any patient. Costs were calculated per patient-quarter, with the surgery date (day 0, index date) placed at the midpoint of quarter 0 (Q0), so Q0 contained costs incurred during the 45.7 days before and after surgery as well as on the surgery date itself.

CRS has traditionally been divided into two main phenotypes, CRS with nasal polyps (CRSwNP) and CRS without nasal polyps, with differences in underlying pathophysiology and association with other conditions such as asthma.[21] Patients with CRSwNP are more likely to have higher disease burden and more likely to receive surgery.[22] Accordingly, participants were split into two subgroups as in our previous work,[13 17 23] according to the patient's polyp status: positive polyp status, where polyps were specifically recorded or implied in the EHR at some point during the patient's follow-up (see online supplemental material, section A); or unknown polyp status, meaning either that polyps were absent or that they were perhaps present but were not recorded.

A flow chart illustrating the relationships between the overall diagnosis cohort, the smaller surgical cohort used in this analysis and the two polyp-based subgroups is given in online supplemental material, section A.

### Resource use and unit costs

Costs were calculated from an NHS perspective,[24] and prices were in 2017–2018 UK pound sterling. Resource use data were extracted on numbers and types of

consultations, investigations, procedures including surgeries, and prescriptions, and classified according to categories available in the relevant published unit costs.

Cost information was categorised for analysis according to these five groups: (1) hospital admitted patient care (APC) from HES APC events (costed as day case or elective inpatient); (2) hospital OP attendances from HES OP events; and (3) primary care visits (GP contacts, practice nurse contacts, other primary care contacts), (4) primary care antibiotic prescriptions, and (5) other relevant primary care prescriptions, with the latter three groups all from CPRD events data.

Inpatient and OP care codes included sinus procedures, nose procedures, nasal polypectomy and diagnostic imaging, and were grouped into cost categories as detailed in online supplemental material, section B, table B1, and NHS reference costs[25] were applied. Inpatient care lasting less than 1 day according to the duration captured in CALIBER was costed as a day case, and stays longer than 1 day were costed as elective inpatient admissions. NHS reference costs from 2017 to 2018 were used where available for that category, or earlier NHS reference costs were used where required, with uplift to 2017–2018 prices using HCHS (Hospital and Community Health Service) inflation indices.[26] This was required for OP complex sinus procedures (2016–2017 prices were used and uplifted) and OP major sinus procedures (2015–2016 prices used and uplifted).

Unit costs and related information for primary care consultations were obtained from the Personal Social Services Research Unit[26 27] (see online supplemental material, section B, table B2). Longitudinal CPRD data which looked at GP contacts in England in 2010–2011 for respiratory tract infections suggested that 1% of adults received treatment for rhinosinusitis from their GP each year, with a median of four GP visits, and with 91% of these patients receiving an antibiotic prescription,[28] so antibiotic prescriptions from primary care were analysed as a separate category. The data set contained six commonly used antibiotics that were costed separately, and 38 less common antibiotics that were grouped together and a mean cost applied. The non-antibiotic medications comprised corticosteroids (including combinations with antihistamine) and all other drugs (ie, painkillers, antihistamines, decongestants and combinations thereof). Unit costs were obtained from the British National Formulary[29] (see online supplemental material, section B, table B3).

## Statistical analysis

Poisson regression was used to calculate incidence rates per quarter (91.3 days) for each of the five types of event listed in the Resource use and unit costs section, split by polyp status, and unit costs described above were applied to event rates to calculate costs.

Events were censored at 10 years before or after the surgery date for inpatient and primary care, and at 2 years before and after for OP care, as including events at dates further away led to small event numbers and therefore large uncertainties (see online supplemental material, section C, table C1, for the denominators at each time-point, ie, numbers of patients at risk of having a health-care event at that moment according to their presence within the follow-up period). The total costs were therefore calculated in the period covering 2 years before and after surgery, split into quarters and also summarised as 1-year costs from surgery to allow comparison with other studies.

Discounting was not included as future costs were not projected. Information from electronic records was considered complete, so no imputation was performed. Stata V.16 was used to run the analyses.[30] Mean per-person-quarter costs split according to the five categories above were calculated for the quarter containing the surgery date at its midpoint (Q0), and the mean per quarter for the eight quarters before and eight quarters after Q0, to provide estimates of costs for surgical patients both in the lead up to their surgery and in subsequent months, as well as around the surgery date itself. Total 1-year surgery costs were also calculated per person by summing the four quarters from surgery, that is, summing costs from Q0, Q1, Q2 and Q3.

## Patient and public involvement

Patient and public involvement collaborators are involved in the MACRO programme including its design, conduct, reporting and dissemination, but were not directly involved in the production of this cost analysis publication.

## RESULTS

### Patient cohort and demographics

Of the 62 685 patients identified as definitely having CRS in 1997–2016 and registered in the GP practices covered by the CPRD in England, 13 462 received CRS-related surgery and were included in this analysis. Two-thirds (9056, 67%) were in the polyp-positive subgroup, with the rest (4406, 33%) in the polyp-unknown subgroup. In the wider group including patients who were CRS definite both with and without surgery (n=62 685), these proportions were reversed, namely one-third (23 036, 37%) were polyp positive and two-thirds (39 649, 63%) were not. These proportions agree with other published work regarding the incidence of nasal polyps in patients with CRS.[22 31–33] Patient demographic information is presented in table 1.

### Total costs

The total per-person costs to the NHS for 1 year (Q0–Q3) in patients receiving surgery for CRS were £1408 in those with unknown polyp status, £2547 in those with known positive polyp status and £2173 overall for all patients. The majority of this expenditure took place in Q0 (table 2) and the highest single cost category was polypectomy in the polyp-positive group (table 3). Table 2 shows the mean per-patient-quarter costs, total and by cost component over the 2-year period before the surgery date,

**Table 1** Patient demographic information at surgery date

| | Unknown polyp status | | Positive polyp status | | All patients | |
|---|---|---|---|---|---|---|
| Total patients, n | 4406 | | 9056 | | 13462 | |
| Age in years, mean (SD) | 42.4 (14.6) | | 47.9 (14.7) | | 46.1 (14.9) | |
| | n | % | n | % | n | % |
| Sex | | | | | | |
| Male | 2029 | 46.1 | 6073 | 67.1 | 8102 | 60.2 |
| Female | 2377 | 53.9 | 2983 | 32.9 | 5360 | 39.8 |
| Ethnicity | | | | | | |
| White | 4038 | 91.6 | 8264 | 91.3 | 12302 | 91.4 |
| India/South Asia | 88 | 2.0 | 209 | 2.3 | 297 | 2.2 |
| Black | 45 | 1.0 | 68 | 0.8 | 113 | 0.8 |
| China/East Asia | 42 | 1.0 | 81 | 0.9 | 123 | 0.9 |
| Mixed | 51 | 1.2 | 120 | 1.3 | 171 | 1.3 |
| Unknown | 142 | 3.2 | 314 | 3.5 | 456 | 3.4 |
| Region of England | | | | | | |
| North East | 51 | 1.2 | 179 | 2.0 | 230 | 1.7 |
| North West | 809 | 18.4 | 1585 | 17.5 | 2394 | 17.8 |
| Yorkshire | 208 | 4.7 | 444 | 4.9 | 652 | 4.8 |
| East Midlands | 126 | 2.9 | 287 | 3.2 | 413 | 3.1 |
| West Midlands | 399 | 9.1 | 1044 | 11.5 | 1443 | 10.7 |
| East | 516 | 11.7 | 1109 | 12.2 | 1625 | 12.1 |
| South West | 627 | 14.2 | 1192 | 13.2 | 1819 | 13.5 |
| South Central | 523 | 11.9 | 953 | 10.5 | 1476 | 11.0 |
| London | 543 | 12.3 | 1072 | 11.8 | 1615 | 12.0 |
| South East | 604 | 13.7 | 1191 | 13.2 | 1795 | 13.3 |

**Table 2** Costs per patient-quarter, broken down by healthcare/prescription category, by time period and by polyp status

| | Inpatient care (DC and EL) | Outpatient | Primary care consultations | Primary care Abx | Primary care non-Abx | Total |
|---|---|---|---|---|---|---|
| Mean per-patient-quarter costs over 2 years before surgery (−Q1 to −Q8) | | | | | | |
| Unknown polyps (£) | 3.35 | 40.83 | 16.08 | 1.70 | 5.87 | 67.82 |
| Positive polyps (£) | 1.53 | 29.69 | 16.68 | 1.15 | 7.64 | 56.69 |
| All patients (£) | 2.13 | 33.49 | 16.49 | 1.33 | 7.06 | 60.50 |
| Per person-quarter (in Q0, containing index surgery) | | | | | | |
| Unknown polyps (£) | 1117.37 | 75.68 | 7.04 | 1.27 | 5.54 | 1206.90 |
| Positive polyps (£) | 2284.63 | 62.41 | 5.59 | 0.99 | 7.79 | 2361.42 |
| All patients (£) | 1902.00 | 66.75 | 6.06 | 1.08 | 7.06 | 1982.95 |
| Mean per-patient-quarter costs over 2 years after surgery (Q1–Q8) | | | | | | |
| Unknown polyps (£) | 8.64 | 37.71 | 6.43 | 1.26 | 5.50 | 59.54 |
| Positive polyps (£) | 20.70 | 25.60 | 4.73 | 0.95 | 7.63 | 59.62 |
| All patients (£) | 16.87 | 29.46 | 5.27 | 1.05 | 6.96 | 59.61 |

Prices in 2017–2018 UK pound sterling.
Abx, antibiotics; DC, day case; EL, elective inpatient.

**Table 3** Mean inpatient costs per patient-quarter in Q0 by procedure category, split by polyp status

| Q0 | CT/other imaging, DC | Minor nose including biopsy, DC | Intermediate nose and minor sinus, DC | Intermediate/major/complex sinus, DC | CT/other imaging, EL | Minor nose including biopsy, EL | Intermediate nose and minor sinus, EL | Intermediate/major/complex sinus, EL | Polypectomy | Total (Q0) |
|---|---|---|---|---|---|---|---|---|---|---|
| Unknown polyps | 0.16 | 8.62 | 43.86 | 243.32 | 0.86 | 29.10 | 110.97 | 680.49 | 0.00 | 1117.37 |
| Positive polyps | 0.02 | 7.85 | 40.09 | 152.60 | 0.14 | 28.44 | 166.57 | 474.22 | 1414.69 | 2284.63 |
| All patients | 0.07 | 8.10 | 41.32 | 181.87 | 0.37 | 28.66 | 148.42 | 540.56 | 952.62 | 1902.00 |

Prices in 2017–2018 UK pound sterling.
CT, computed tomography; DC, daycase; EL, elective inpatient; Q0, quarter containing surgery date at centre.

during Q0 when surgery took place and over the 2-year period after surgery. Inpatient care costs peaked during Q0 and comprised the majority of Q0 costs. OP costs during Q0 were approximately twice those in the before or after periods but small in comparison to Q0 inpatient costs. The cost of primary care consultations appeared to be lower during Q0 compared with the time preceding surgery and did not rebound in the following 2 years, and the two categories of primary care prescription costs were low at all times, with little apparent change around the surgery date. The SEs for the mean per-patient-quarter costs in the 2 years before and after surgery are given in table 4 but are omitted from table 2 for readability purposes.

### APC: day case (<1 day) and elective inpatient (>1 day)

Hospital admission costs were £2.13 (SE £1.18) per patient-quarter in the eight quarters leading up to the surgery quarter (£1.53 (SE £0.93) in polyp-positive patients and £3.35 (SE £2.11) in polyp-unknown patients) (see table 4). The majority of these costs were during Q0 (£1902 overall; £1117 in polyp-unknown patients and £2285 in polyp-positive patients), and costs per patient-quarter in the subsequent eight quarters were lower than this peak, at £16.87 (SE £2.97) per patient-quarter (see table 4).

Regarding revision surgeries, 0.4% of patients in this analysis had a second surgery during the second half of Q0 after their index surgery, and 4.9% of patients received a second surgery at some point during the eight quarters following Q0. These subsequent surgeries were identified using the same codes as those by which the patients were selected into the cohort, and were included in the costs simply as downstream hospital costs. There was no evidence of a preferred length of wait between first and second surgeries.

Table 3 shows the cost breakdown during Q0. The highest expenditure in polyp-positive patients was on polypectomy (E081), covering around one-third of all events in this group, and a further 40% corresponded to one of functional endoscopic sinus surgery (FESS), intranasal antrostomy or intranasal ethmoidectomy, which together formed the major part of the intermediate/major/complex sinus procedure group. In polyp-unknown patients, the highest expenditure was on FESS, intranasal antrostomy or intranasal ethmoidectomy, which again formed the major part of the intermediate/major/complex sinus procedure group. Types of procedures were grouped together as seen in table 3 as some codes had small event numbers, thus regressions did not converge unless some groupings were made beyond the categories listed in online supplemental material, section B, table B1. Groupings were made based on consecutive unit costs in elective inpatient data and the same groupings were used in day case data for consistency of reporting. Tables showing costs split by category and polyp subgroup are given in the online supplemental material, section D.

**Table 4** Costs during the surgery quarter (Q0) and 2 years before and after

| | Per-patient costs over 2 years preceding surgery | Mean (SE) per person-quarter over 2 years preceding surgery | Per-patient costs in the quarter containing surgery (Q0) | Mean (SE) per person-quarter over 2 years following surgery | Per-patient costs over 2 years following surgery |
|---|---|---|---|---|---|
| Inpatient costs (DC and EL) | | | | | |
| Unknown polyps | 26.81 | 3.35 (2.11) | 1117.37 | 8.64 (2.97) | 69.15 |
| Positive polyps | 12.26 | 1.53 (0.93) | 2284.63 | 20.70 (4.56) | 165.61 |
| All patients | 17.02 | 2.13 (1.18) | 1902.00 | 16.87 (2.97) | 134.96 |
| Outpatient costs | | | | | |
| Unknown polyps | 326.61 | 40.83 (12.22) | 75.68 | 37.71 (8.40) | 301.69 |
| Positive polyps | 237.49 | 29.69 (11.41) | 62.41 | 25.60 (4.64) | 204.77 |
| All patients | 267.93 | 33.49 (11.57) | 66.75 | 29.46 (5.78) | 235.67 |
| Primary care consultations | | | | | |
| Unknown polyps | 128.64 | 16.08 (5.09) | 7.04 | 6.43 (0.62) | 51.47 |
| Positive polyps | 133.48 | 16.68 (7.02) | 5.59 | 4.73 (0.16) | 37.87 |
| All patients | 131.91 | 16.49 (6.28) | 6.06 | 5.27 (0.28) | 42.18 |
| Primary care antibiotic prescriptions | | | | | |
| Unknown polyps | 13.57 | 1.70 (0.35) | 1.27 | 1.26 (0.04) | 10.05 |
| Positive polyps | 9.20 | 1.15 (0.20) | 0.99 | 0.95 (0.03) | 7.60 |
| All patients | 10.63 | 1.33 (0.24) | 1.08 | 1.05 (0.02) | 8.38 |
| Primary care non-antibiotic prescriptions | | | | | |
| Unknown polyps | 46.93 | 5.87 (0.80) | 5.54 | 5.50 (0.09) | 43.96 |
| Positive polyps | 61.12 | 7.64 (1.25) | 7.79 | 7.63 (0.07) | 61.08 |
| All patients | 56.48 | 7.06 (1.10) | 7.06 | 6.96 (0.05) | 55.65 |

Prices in 2017–2018 UK pound sterling.

DC, day case; EL, elective inpatient; Q0, quarter containing surgery date at centre; SE, standard error.

### OP attendances

OP care costs were £33.49 (SE £11.57) per patient-quarter in the eight quarters preceding surgery (£29.69 (SE £11.41) in polyp-positive patients and £40.83 (SE £12.22) in polyp-unknown patients) (see table 4), then £66.75 during Q0 (£62.41 for polyp positive and £75.68 for polyp unknown). Costs per patient-quarter were reduced from this peak in the subsequent eight quarters, at around £30 per patient-quarter (see table 4).

Table 5 shows the breakdown of costs during Q0 and the quarters immediately preceding and succeeding Q0. The highest expenditure in both subgroups was on CT/other scans, which comprised around two-thirds of CT scans and one-third X-rays. All categories showed a peak in costs in Q0 except for CT/other scans, which instead had a slightly higher peak in the quarter immediately preceding surgery (see table 5). This tallies with the advice in EPOS (European Position Paper on Rhinosinusitis and Nasal Polyps) 2020 stating that CT scans should always be given before surgery.[1] Tables showing the values split by category and by polyp subgroup, and graphs illustrating this information (ie, expanding on information in table 5) are given in online supplemental material, section E.

### Primary care consultations

Primary care consultation costs were £16.49 (SE £6.28) per patient-quarter in the eight quarters preceding surgery (£16.68 (SE £7.02) in polyp-positive patients and £16.08 (SE £5.09) in polyp-unknown patients) (see table 4), then £6.06 during Q0 (£7.04 in polyp-unknown patients, £5.59 in polyp-positive patients), and costs per patient-quarter were similarly reduced in the subsequent eight quarters, at around £5–£6 per patient-quarter (see table 4). The highest expenditure in both subgroups was GP face-to-face consultations at the GP practice. Tables showing the values split by category and by polyp subgroup, and

**Table 5** Mean outpatient costs per person-quarter in Q0 and the immediately preceding and succeeding quarters, by procedure category, split by polyp status

| | CT/other imaging | Minor nose including biopsy | Intermediate nose and minor sinus | Intermediate sinus | Major/complex sinus | Polypectomy | Total (by person-quarter) |
|---|---|---|---|---|---|---|---|
| Polyps unknown | | | | | | | |
| −Q1 | 32.30 | 4.55 | 11.26 | 14.70 | 5.18 | – | 67.99 |
| Q0 | 29.11 | 4.37 | 13.95 | 17.65 | 10.59 | – | 75.68 |
| Q1 | 25.51 | 3.16 | 12.17 | 12.35 | 5.51 | – | 58.70 |
| Polyps positive | | | | | | | |
| −Q1 | 25.04 | 1.84 | 11.36 | 12.02 | 6.17 | 0.44 | 56.87 |
| Q0 | 23.03 | 2.93 | 12.47 | 14.46 | 8.66 | 0.85 | 62.41 |
| Q1 | 14.88 | 1.63 | 7.57 | 8.64 | 2.97 | 0.32 | 36.00 |
| All patients | | | | | | | |
| −Q1 | 27.49 | 2.76 | 11.33 | 12.93 | 5.83 | 0.29 | 60.62 |
| Q0 | 25.01 | 3.40 | 12.96 | 15.51 | 9.29 | 0.57 | 66.75 |
| Q1 | 18.32 | 2.13 | 9.06 | 9.84 | 3.79 | 0.21 | 43.35 |

Prices in 2017–2018 UK pound sterling.
CT, computed tomography; Q0, quarter containing surgery date at centre.

graphs illustrating this information, are given in online supplemental material, section F.

### Primary care prescriptions: antibiotics

Primary care antibiotic prescription costs were £1.33 (SE £0.24) per patient-quarter in the eight quarters before surgery (£1.15 (SE £0.20) in polyp-positive patients and £1.70 (SE £0.35) in polyp-unknown patients), then £1.08 during Q0 (£1.27 in polyp-unknown patients, £0.99 in polyp-positive patients), and similar in the subsequent eight quarters, at around £1 per patient-quarter (see table 4). The highest expenditure was on tetracyclines, followed by macrolides, and tables showing the values split by category and by polyp subgroup, and graphs illustrating this information, are shown in the online supplemental material, section G.

### Primary care prescriptions: steroids and other non-antibiotics

Primary care non-antibiotic prescription costs were primarily for corticosteroids, plus general sinusitis drugs like painkillers and decongestants, and were £7.06 (SE £1.10) per patient-quarter in the eight quarters before surgery (£7.64 (SE £1.25) in polyp-positive patients and £5.87 (SE £0.80) in polyp-unknown patients), then £7.06 during Q0 (£7.79 for polyp unknown, £5.54 for polyp positive), and similar in the subsequent eight quarters, at around £7 per patient-quarter (see table 4). Tables showing the values split by category and polyp subgroup, and graphs illustrating this information, are given in online supplemental material, section H. This information includes only prescriptions made by the GP, and does not include other medications bought over the counter by the patient.

## DISCUSSION

In this paper, we have shown that inpatient surgical sinus procedures and nasal polypectomies are the largest healthcare cost in patients receiving surgery for CRS when considering the costs of primary and secondary care to the NHS in England, at around £1000–£2000 per person-quarter in the quarter containing the surgery date (Q0). Other secondary and primary healthcare costs in the eight quarters before and after Q0 are considerably smaller, at around £60 per person-quarter across polyp subgroups. These are average values over the whole population and are not split according to demographic groups.

Average total costs across secondary and primary care settings were £1983 per patient overall during Q0, or £2361 per polyp-positive patient and £1207 per polyp-unknown patient, in 2017–2018 prices. Hospital overnight admission and day case inpatient costs incurred during Q0 were the costliest category across the 4.25-year analysis period, dwarfing other cost components. Primary care prescription costs were low across both groups, with antibiotics costing around £1 per person-quarter and non-antibiotics around £7 per person-quarter. OP care costs appeared higher than primary care costs at around £30 per person-quarter before and after surgery, and around £67 per person-quarter during Q0. Primary care consultation costs appeared higher before surgery than after (£16 vs £6 per person-quarter), and inpatient care costs appeared higher after surgery than before (£17 vs £2 per person-quarter). These findings suggest that the costs to the NHS associated with CRS, especially the non-surgical costs, are currently low. They also suggest that CRS surgery does not appreciably impact overall management costs, either upwards or downwards, although these

costs are low so it would be difficult to see a meaningful change. These values are presented as descriptive statistics and formal significance testing among the various categories and timepoints described above has not been performed.

There were certain limitations in this analysis. Only costs for those patients for whom CRS surgery codes were recorded during the time period were included, and the analysis was based around the date of their first CRS surgery as captured during the analysis time period. If a patient had another surgery before they entered the cohort, this would not have appeared in the data set, thus we cannot be entirely certain that the index surgery was indeed the patient's first CRS surgery.

Other limitations relate to other aspects of coding and identification of patients and their treatments, as the data set used was collected by hospitals and GP practices for reimbursement and clinical management purposes, and not specifically for research purposes, and patients were not prospectively recruited into the data set so there was no prospectively defined baseline. For example, identification of patients with CRS and their diagnosis dates and treatment information was performed using phenotyping code lists of treatments and diagnostic markers, using methodology common to observational analyses using routine data and expert clinical opinion to determine the code lists. Thus, the identification of patients and treatments was reliant on patients' practitioners or coding staff having entered certain codes or combinations of codes. Furthermore, the coding regarding polyp status is limited, as there is no code to confirm that a patient does not have polyps, there is only the absence of a positive report of polyps. This is based on treatments recorded, including the reporting of a polypectomy, leading to a certain circularity when reporting the treatments received by subgroup.

This analysis used CPRD for primary care information, which covers around 8% of the population of England, and is broadly representative of the UK although with acknowledged gaps including people who are universally under-represented in UK healthcare systems, for example, homeless people and those with non-standard residency or migration status.[34]

We used the standard English NHS cost perspective, although we did not have information on Personal Social Services, the costs of which would normally be included in analyses for the National Institute for Health and Care Excellence,[24] or on other community-based healthcare such as Improving Access to Psychological Therapies, which might be relevant to this population. We also did not have emergency care costs, but we do not anticipate that this would be a major part of this care pathway. We had no information on wider societal costs, for example, relating to productivity (time off work) or any out-of-pocket costs for patients. It is possible therefore that information regarding factors that are important to patients and their families was not captured in this analysis.

Other work published in this area has focused mostly on US costs and used different unit costs and included different cost categories. Bhattacharyya et al[35] investigated the costs of patients with CRSwNP in a US claims database using information gathered in 2013–2014, beginning at CRS diagnosis. When patients with CRSwNP undergoing FESS were compared with patients with CRSwNP not undergoing surgery, they found that the extra cost of surgery during that first year was $13 532. This was an observational, retrospective case–control study, meaning that treatment decisions were not randomly assigned within the CRSwNP group, and therefore any differences in costs according to treatment decisions were susceptible to selection bias. Studies have also been published examining cost breakdowns of patients with CRS in the USA regarding the distribution of expenditure across different care categories. For example, Caulley et al[36] considered all patients with CRS in the US Medical Expenditure Panel Survey, taking a cross section in 2011, and found that ambulatory office-based consultations and prescriptions each accounted for a greater proportion of expenditure than inpatient hospital visits, although this was for all patients with CRS, not just those receiving surgery, and the US system is both structured and financed quite differently from the UK system. For example, certain medications available in North America for the management of CRSwNP like monoclonal antibodies are not available in the English NHS, and therefore no patient in the present analysis had received these. Aspirin desensitisation also has very restricted availability in the UK and is only offered in a small number of UK centres so was also not captured here. Bhattacharyya et al[35] however reported that prescription costs were not a major part of CRS costs for patients with CRS undergoing surgery or not undergoing surgery in their observational study using the Truven Health MarketScan US claims database.

Our analysis only included surgical patients with CRS, and did not attempt to include non-surgical patients to allow comparison of treatments received by surgical and non-surgical patients, as this is difficult to do in observational data sets and can lead to misleading results, with important limitations due to the lack of randomisation, as there are unobserved and unmeasured confounders that can govern what treatment people receive. RCTs aim to identify and capture these confounders, using a large enough sample size that there is balance across the arms, and the analysis is adjusted for confounders. There are methods such as instrumental variable analysis that attempt to mimic randomisation using statistical methods, but it is typically hard to find a suitable instrument.[37 38] Using random allocation to assign treatments is therefore a powerful tool in eliminating selection bias, and is not available in analysis using routine observational data, hence the importance of the MACRO RCT,[12] which began recruiting patients in 2018. MACRO is randomising patients 1:1:1 to receive appropriate medical therapy (AMT), surgery plus AMT or long-term low-dose macrolides plus AMT, and collecting all relevant information

required to make a randomised comparison between surgery and non-surgical treatments in a full cost-utility analysis.[39–41] The MACRO RCT will provide key information regarding changes in QOL on receiving surgery for CRS and allow us to provide information regarding the relative cost-effectiveness of surgery and other treatments in the UK context.

## CONCLUSION

This is the first study we are aware of that analysed the costs of primary and secondary healthcare received by patients undergoing surgery for CRS using English NHS costs. It included a large sample size that was representative of care given by the NHS in England and showed that the inpatient costs including CRS surgery itself were around £2000 during the quarter containing surgery, and that the cost of management before and after surgery in primary and secondary care settings was low in comparison at around £60 per person-quarter in the two preceding and subsequent years.

This study reports important new evidence regarding the cost of English NHS healthcare costs for patients receiving CRS surgery, and provides further justification for the use of randomised clinical trials to investigate the relative cost-effectiveness of surgical treatments for CRS, as well as providing useful information that can be applied in future work in the UK and similar contexts, including our own future analysis of the MACRO trial data.

**Author affiliations**
[1]Research Department of Primary Care and Population Health, University College London, London, UK
[2]Department of Medical Statistics, London School of Hygiene & Tropical Medicine, London, UK
[3]Health Data Research UK, London, UK
[4]Institute of Health Informatics, University College London, London, UK
[5]MRC Clinical Trials Unit at UCL, University College London, London, UK
[6]Primary Care and Population Science, University of Southampton, Southampton, UK
[7]ENT Clinical, Zeist, The Netherlands
[8]NIHR UCLH Biomedical Research Centre, London, UK
[9]Ear Institute, University College London, London, UK
[10]Norwich Medical School, University of East Anglia, Norwich, UK
[11]ENT Department, James Paget University Hospitals NHS Foundation Trust, Great Yarmouth, UK
[12]ENT Department, Guy's and St Thomas' NHS Foundation Trust, London, UK
[13]Primary Care Unit, Department of Public Health and Primary Care, University of Cambridge, Cambridge, UK

**Acknowledgements** The authors wish to acknowledge the valuable input from the MACRO WS1 study team and the wider MACRO programme team, including Hannah Evans, and our Patient and Public Involvement collaborators.

**Contributors** CSC was the lead author and wrote the first draft of the article and is responsible for the overall content as guarantor. CSC and EW cleaned and analysed the data regarding rates of events. CSC and SM planned and conducted the cost analyses. EW, CSC, SM, MT, CH and CMP formulated the phenotyping code lists for identifying the patient cohort and relevant treatments and diagnoses. CSC, EW, SD, JRC, MT, HB, AGMS, CMP, CH and SM were involved in formulating the overall research question, and in designing and conducting the study. All authors contributed to and approved the final manuscript.

**Funding** This work was supported by the National Institute for Health Research (NIHR) under its Programme Grants for Applied Research (PGfAR) programme (grant reference number: RP-PG-0614-20011). SD is supported by (1) the NIHR

University College London Hospitals Biomedical Research Centre (UCLH BRC); (2) Health Data Research UK (award reference: LOND1), which is funded by the UK Medical Research Council, Engineering and Physical Sciences Research Council, Economic and Social Research Council, Department of Health and Social Care (England), Chief Scientist Office of the Scottish Government Health and Social Care Directorates, Health and Social Care Research and Development Division (Welsh Government), Public Health Agency (Northern Ireland), British Heart Foundation and Wellcome Trust; (3) BigData@Heart programme that has received funding from the Innovative Medicines Initiative 2 Joint Undertaking under grant agreement number 116074 (this Joint Undertaking receives support from the European Union's Horizon 2020 research and innovation programme and EFPIA); and (4) the British Heart Foundation Accelerator Award (AA/18/6/24223).

**Disclaimer** The views expressed are those of the author(s) and not necessarily those of the National Health Service, the National Institute for Health Research or the Department of Health and Social Care.

**Competing interests** EW: personal fees from AstraZeneca for provision of training on propensity score methodology unrelated to the current manuscript. JRC: grants from the UK Medical Research Council (grant numbers MC_UU_12023/21 and MC_UU_12023/29), consultancy from AstraZeneca and Novartis on statistical methodology for the analysis of partially observed data, and book royalties from 'Multiple Imputation and its Application' (Wiley) and 'Meta-analysis using R' (Springer). CMP: personal fees for advisory work from GSK and Sanofi, and work as a Trustee of Fifth Sense. CH: advisory board for GSK, Sanofi and AstraZeneca and speaker fees for Mylan and Intersect.

**Patient consent for publication** Not required.

**Ethics approval** This study used routinely collected data. Scientific and ethical approval for the use of and data linkages within the Clinical Practice Research Datalink (CPRD) primary care data was obtained following application to the Independent Scientific Advisory Committee (ISAC), a non-statutory expert advisory body (protocol number: 16_200).

**Provenance and peer review** Not commissioned; externally peer reviewed.

**Data availability statement** Data may be obtained from a third party and are not publicly available. The data were pseudonymised patient-level data from the CALIBER resource and are not publicly available. Analysis code can be made available on reasonable request and in accordance with relevant guidance.

**ORCID iDs**
Caroline S Clarke http://orcid.org/0000-0002-4676-1257
Carl M Philpott http://orcid.org/0000-0002-1125-3236
Claire Hopkins http://orcid.org/0000-0003-3993-1569
Stephen Morris http://orcid.org/0000-0002-5828-3563

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
