## [Reviewer comments · BMJ Open]

ARTICLE DETAILS

TITLE (PROVISIONAL)	Observational retrospective study calculating health service costs of patients receiving surgery for chronic rhinosinusitis in England, using linked patient-level primary and secondary care electronic data
AUTHORS	Clarke, Caroline; Williamson, Elizabeth; Denaxas, S; Carpenter, James; Thomas, Mike; Blackshaw, Helen; Schilder, Anne; Philpott, Carl; Hopkins, Claire; Morris, Stephen

VERSION 1 – REVIEW

REVIEWER	Thavorn, Kednapa Institute for Clinical Evaluative Sciences, ICES @uOttawa
REVIEW RETURNED	12-Oct-2021

GENERAL COMMENTS	Based on the linked electronic health records in England, the authors estimated the health care cost of chronic rhinosinusitis (CRS) patients who received surgery from the National Health Service (NHS) perspective. The study showed that surgery for CRS cost the NHS £2,173 per year; inpatient costs were the key driver of the total costs per person per quarter. The study has the potential to provide helpful cost estimates for future economic evaluations of health care interventions for patients with CRS. However, it is unclear whether the estimated costs were attributable to CRS surgery because several factors, such as patient demographic (age, sex, geographical residence) and clinical (comorbidities), may confound the association between CRS surgery and health care costs. It is also unclear why the authors calculated the costs 2 years before and after surgery. Patients might experience other medical conditions unrelated to CRS during such periods, which might overestimate the total costs of CRS. In addition, the study included CRS patients and health care utilization from 1997 to 2016, covering ~20 years of data. It is questionable whether the use of health services and costs estimated in the past decade represent the current clinical practice for CRS. Another major question is related to the diagnostic accuracy of codes used to identify CRS patients. Have the diagnostic and procedure codes been validated? What was the purpose of a Poisson regression? Was it used for the adjustment for confounding factors? Was it performed on each of the health care services? It would be helpful to include a flow chart describing how participants were identified and included in the study.
--

REVIEWER	Yong, Michael The University of British Columbia, Otolaryngology - Head and Neck Surgery
REVIEW RETURNED	15-Oct-2021

GENERAL COMMENTS	Thank you for this thorough review of the costs related to CRS in the UK. It is an important addition to the literature which will allow further subsequent research. The methodology is well-done and the discussion is appropriate. I was wondering if you might add, as an additional issue in the limitations in the discussion section, the absence of included costs related to more severe forms of CRS like AERD, and how medical therapies such as mono-clonal antibodies and allergy/asthma-related costs are not included in this analysis. Since it is known that about 10% of patients with CRSwNP have AERD, additional costs related to aspirin desensitization or mono-clonal antibody medical therapy such as with dupilumab may be sizeable, but do not seem to be captured in this study. Given the nature of the dataset used, it is understandable that these costs are not captured - however, it may be helpful to mention these gaps in the discussion. In addition, I was wondering about the costs that do not seem to be captured regarding specialist visits. Reading through the article, it seems that costs of healthcare visits are only associated with primary care. However, I assume that when a patient receives a surgery, they will follow-up with a specialist for surgery-related follow-up and complications. Was this included in the analysis?
---

REVIEWER	Gill, Amarbir The University of Utah School of Medicine
REVIEW RETURNED	21-Oct-2021

GENERAL COMMENTS	The authors should be commended on their investigation focused on research area that has no data to date specific to the UK and thus represents a critical knowledge gap. Overall, this is a very well-written manuscript; it is clear, concise, and flows well. It represents a good first step in identifying potential variables that should be further considered when analyzing cost and management of CRS. I have a few concerns/suggestions that are outlined below.  1. Consider shortening/combining the first two paragraphs of the introduction as there is information here that is not critical to the narrative. This would allow the reader to get to the third paragraph sooner, which does a great job of setting up the question/objective of the study and how it is going to be answered in the rest of the paper 2. Within the limitations, please also acknowledge a potential for error in selection of procedure codes for correctly capturing surgeries for CRS only. For example, inpatient costs are much higher in the “unknown polyp” group compared to the “polyp” group – this observation calls into question whether non-CRS surgeries (ie skull base resections) were potentially included in the procedures done in the “unknown polyp,” as inpatient stays are more likely in skull base resections. I understand this may not be
--

	easy to uncover, but it should at least be acknowledged as a limitation of the study. 3. How did the study address potential revision surgeries that occurred during the f/u period? This is briefly alluded to in the discussion section but would be better placed in the methods section. For example, when revision surgeries did occur, were they simply overlooked/ignored in calculating costs? 5% (those that needed revision surgery during the time period) is not insignificant if it is not being incorporated into costs. This is a limitation of the study, as it prevents the study from portraying the true cost of CRS care. If not able to incorporate the costs of the revision surgeries, it may be advisable to remove all patients who had revision surgery and only include those who required one surgery during the time period analyzed (this would allow for more homogeneity within the analyses). 4. The importance of separate analyses focusing on unknown polyp patients are difficult to interpret, given the ambiguity surrounding which patients may actually be present in this cohort. Consider having a polyp only cohort and a total cohort, and removing unknown polyp cohort, or otherwise provide justification for why it is important to consider the unknown polyp cohort separately, what this analysis adds, and how to interpret it given the ambiguity of its definition. 5. In the supplementary document, section D – centered is misspelled centred in the table title.
--	---

VERSION 1 – AUTHOR RESPONSE

Reviewer 1	The study has the potential to provide helpful cost estimates for future economic evaluations of health care interventions for patients with CRS. However, it is unclear whether the estimated costs were attributable to CRS surgery because several factors, such as patient demographic (age, sex, geographical	Thanks for this comment. We note that this cost analysis did not cover all CRS patients, instead it specifically covered only those CRS patients who had had CRS surgery, so the analysis cohort was defined on the basis of having received this type of surgery (within acknowledged coding limitations). We were not looking for an association between having surgery and any cost variable as they had all had surgery. The costs that were calculated for each patient were health care costs, and were found by applying a standard unit cost to the reported health care resource use events (where an event was an appointment, prescription, or surgery, etc.). We have added some words in
---	--

	residence) and clinical (comorbidities), may confound the association between CRS surgery and health care costs.	the Abstract and Methods to clarify this.	
4	Reviewer 1	It is also unclear why the authors calculated the costs 2 years before and after surgery. Patients might experience other medical conditions unrelated to CRS during such periods, which might overestimate the total costs of CRS.	Thanks for this comment. The reasons for the durations chosen are described in section 2.3. It is possible that some of the events could have been for conditions other than CRS, although this was not thought likely on consideration of the types of events that were included, as they were all potentially related to CRS (see section B of supplementary materials). We have added some text in the limitations about this, around coding for treatments.
5	Reviewer 1	In addition, the study included CRS patients and health care utilization from 1997 to 2016, covering ~20 years of data. It is questionable whether the use of health services and costs estimated in the past decade represent the current clinical practice for CRS.	Thanks for this comment. The reviewer is right of course that practices and treatment pathways do indeed change over time, and treatment pathways have also changed perhaps more than usual over the course of the last couple of years as a result of the Covid-19 pandemic. However, we still consider it to be useful to report these costs, as there is currently very limited information of this type available for this disease and context.
6	Reviewer 1	Another major question is related to the diagnostic accuracy of codes used to identify CRS patients. Have the diagnostic and procedure codes been validated?	Thanks very much for this comment. We did not explicitly validate our own coding scheme, but we used a similar approach to Rudmik et al. and we have clarified this in the Methods and added more references around this.

7	Reviewer 1	What was the purpose of a Poisson regression? Was it used for the adjustment for confounding factors? Was it performed on each of the health care services?	Thanks for this comment. The purpose of the Poisson regression was to obtain estimated rates and 95% confidence intervals for service use, and yes it was used for each of the services, and this latter point has been clarified in section 2.3. Regarding the query about “confounding factors”, the authors note that we were not estimating a treatment effect as such, so there would be nowhere to explore the confounding of any treatment effect by way of some intermediary/confounding variable. It could be that the reviewer is suggesting that the service use costs might vary by demographic group, which is a possibility and is something that could potentially be explored in future work. We were interested here in calculating a mean overall population cost, and have not split this by e.g. age group or sex. The population that we were working with comes from data that are considered to be representative of the population at large, and so we consider that the mean costs we have calculated are appropriate for the population in England. This point is mentioned in the Discussion and a note has been added in the first paragraph of the Discussion for further clarity. We have also modified the third bullet point in the Article Summary to remove any hint or suggestion of a comparison potentially being made, as no comparison was made, and no treatment effect was estimated,
---	------------	--	--

			and we hope that this helps to clarify these issues.
8	Reviewer 1	It would be helpful to include a flow chart describing how participants were identified and included in the study.	Participants were identified using combinations of the code lists that are given in the Appendix. We have constructed a flow chart as suggested and added this to the Supplementary Materials (referring to it in section 2.1 of the manuscript), thanks very much for the suggestion.
9	Reviewer 2	I was wondering if you might add, as an additional issue in the limitations in the discussion section, the absence of included costs related to more severe forms of CRS like AERD, and how medical therapies such as monoclonal antibodies and allergy/asthma-related costs are not included in this analysis. Since it is known that about 10% of patients with CRSwNP have AERD, additional costs related to aspirin desensitization or monoclonal antibody medical therapy such as with dupilumab may be sizeable, but do not seem to be captured in this study. Given the nature of the dataset used, it is understandable that these costs are not captured - however, it may be helpful to mention these gaps in the discussion.	Thanks for this interesting suggestion that addresses the generalisability of our work outside the UK context. Monoclonal antibodies are not available in the English NHS for the management of CRSwNP, and therefore no patient in our study will have received dupilumab or any other mAbs. Aspirin desensitisation has very restricted availability in the UK and is only offered in a small number of centres (Guy's, Wythenshaw and a few others) so will not have been captured, but applies to only very small numbers. We have added some text on this in the Discussion, thanks.
10	Reviewer 2	In addition, I was wondering about the costs that do not seem to be captured regarding specialist visits. Reading through the article, it seems that costs of healthcare visits are only associated with primary care. However, I assume that when a patient receives a surgery, they will follow-up with a specialist for surgery-related follow-up and complications. Was this included in the analysis?	Yes, hospital care of this type was also included (this is referred to in the manuscript as inpatient and outpatient secondary care), and specialist visits would be included in here. We have adjusted some of the wording around hospital/health service care to hopefully clarify this for an international audience.
11	Reviewer 3	Consider shortening/combining the first two paragraphs of the introduction as there is	Thanks for this helpful comment, we have re-

		information here that is not critical to the narrative. This would allow the reader to get to the third paragraph sooner, which does a great job of setting up the question/objective of the study and how it is going to be answered in the rest of the paper	worded in the Introduction as suggested.
12	Reviewer 3	Within the limitations, please also acknowledge a potential for error in selection of procedure codes for correctly capturing surgeries for CRS only. For example, inpatient costs are much higher in the “unknown polyp” group compared to the “polyp” group – this observation calls into question whether non-CRS surgeries (ie skull base resections) were potentially included in the procedures done in the “unknown polyp,” as inpatient stays are more likely in skull base resections. I understand this may not be easy to uncover, but it should at least be acknowledged as a limitation of the study.	Thanks for this comment. There is indeed the potential for coding errors, and limitations regarding coding difficulties and possible errors are discussed in the Discussion section, and some additions have been made here. Procedures that were included are listed in the Supplementary Materials. We note however that the inpatient (day case and elective surgery) costs are in fact higher in the positive polyps group (£2284.63) than in the unknown polyps group (£1117.37) as seen in Table 2.
13	Reviewer 3	How did the study address potential revision surgeries that occurred during the f/u period? This is briefly alluded to in the discussion section but would be better placed in the methods section. For example, when revision surgeries did occur, were they simply overlooked/ignored in calculating costs? 5% (those that needed revision surgery during the time period) is not insignificant if it is not being incorporated into costs. This is a limitation of the study, as it prevents the study from portraying the true cost of CRS care. If not able to incorporate the costs of the revision surgeries, it may be advisable to remove all patients who had revision surgery and only include those who required one surgery during the time period analyzed (this would allow	Revision surgeries would have appeared in the data after the initial surgery, but were not specifically extracted and labelled as revision surgery, rather they were just included as downstream inpatient costs. The codes used to identify and cost revision surgery would have been the same surgery codes as those for which the patients were selected into the cohort, and as such all events within the time period with those codes would have been identified and included in the costs. There was no obvious peak in the inpatient costs at a time after surgery, which suggested that there was no preferred timing for any subsequent surgery, so any revision surgeries would have been

		for more homogeneity within the analyses).	spread out over the time period after the surgery date, as expected. We have added some words in the Results and Discussion around this. The comment in the Discussion was aiming to explain that if a patient had their first surgery before 1997, followed by a revision surgery during our time period of 1997-2016, then we would only have found their revision surgery and would unknowingly have considered that to be their first surgery. This has been moved to the Results section (section 3.3) and some extra words have been included there to clarify this.
14	Reviewer 3	The importance of separate analyses focusing of unknown polyp patients are difficult to interpret, given the ambiguity surrounding which patients may actually be present in this cohort. Consider having a polyp only cohort and a total cohort, and removing unknown polyp cohort, or otherwise provide justification for why it is important to consider the unknown polyp cohort separately, what this analysis adds, and how to interpret it given the ambiguity of its definition	Thanks for this comment. It illustrates the difficulty of using observational data instead of prospectively collected trial data. If we were in a trial (indeed as we are in the MACRO RCT), we would ask all patients to be screened to see definitively whether or not they had polyps, but in observational data we are not routinely told that patients definitely do not have polyps, there is merely the absence of a polyp code or polypectomy code (see code list in the supplementary materials). The best approximation is to say that if there was no positive reporting of polyps, then either the patient didn't have any or they were too small to warrant investigation, hence using the term 'unknown-polyps' instead of 'without-polyps' for the remaining patients who do not fall into the positive-polyps group. Some additional references to our earlier work where this categorisation has been used have been added in

		Section 2.1, and it has been added in the Article Summary bullets as well. The positive polyp group contains CRS patients who definitely had had polyps at some point (notwithstanding coding errors), and the unknown polyp group contains CRS patients who either have no polyps or had very small ones that never gave cause for the doctor to look specifically and find them (or remove them), and therefore no polyps were recorded. The prognosis in the latter group is homogeneous. This is however a limitation of the analysis, and we discuss this in the Discussion section. The results are presented throughout as positive-polyps, unknown-polyps, and total cohort, and we would rather not remove information on the unknown-polyps group as that would seem to leave a gap. (Incidentally, we wonder if these reviewer's comments may have arisen in part due to geographical differences in the use of terminology. In Europe, CRS is used to describe both CRS with and without polyps (CRSwNP and CRSsNP, respectively), while in North America perhaps they are seen and therefore coded as more distinct entities, where CRS is assumed to be CRS without NP, and having nasal polyps (NP) is not necessarily referred to as CRS with NP, rather is referred to as NP alone?)
--	--	--

15	Reviewer 3	In the supplementary document, section D – centered is misspelled centred in the table title.	Thanks for noting this. We have been using UK spelling throughout, so would prefer to keep 'centred' if the editor is happy with this.
----	------------	---	--

VERSION 2 – REVIEW

REVIEWER	Yong, Michael The University of British Columbia, Otolaryngology - Head and Neck Surgery
REVIEW RETURNED	19-Dec-2021

GENERAL COMMENTS	Great edits, covers ground that was unaddressed in the previous draft. The edits in the discussion especially cover some of the limitations of the review well in order to guide the audience as to the generalizability of the data with regards to the rapidly-changing landscape of the subject.
---

REVIEWER	Gill, Amarbir The University of Utah School of Medicine
REVIEW RETURNED	17-Dec-2021

GENERAL COMMENTS	I would like to thank the authors for their clarifications and edits. I think the paper fills a knowledge gap in the literature - congratulations. I have no further comments.
--